# Risk of Early Childhood Obesity in Offspring of Women with Preeclampsia: A Population-Based Study

**DOI:** 10.3390/jcm10163758

**Published:** 2021-08-23

**Authors:** Seung-Woo Yang, Min-Jeong Oh, Keon-Vin Park, Sung-Won Han, Hee-Sun Kim, In-Sook Sohn, Han-Sung Kwon, Geum-Joon Cho, Han-Sung Hwang

**Affiliations:** 1Department of Obstetrics and Gynecology, Sanggye Paik Hospital, School of Medicine, Inje University, Seoul 01757, Korea; mdmichaelyang@paik.ac.kr; 2Department of Obstetrics and Gynecology, College of Medicine, Korea University, Seoul 02841, Korea; mjohmd@korea.ac.kr; 3School of Industrial Management Engineering, Korea University, Seoul 02841, Korea; kbpark16@daum.net (K.-V.P.); swhan@korea.ac.kr (S.-W.H.); 4Department of Obstetrics and Gynecology, Dongguk University Ilsan Hospital, Dongguk University, Goyang 10326, Korea; smallkong7@gmail.com; 5Division of Maternal and Fetal Medicine, Department of Obstetrics and Gynecology, Research Institute of Medical Science, School of Medicine, Konkuk University, Seoul 05030, Korea; 19960011@kuh.ac.kr (I.-S.S.); 20080251@kuh.ac.kr (H.-S.K.)

**Keywords:** preeclampsia, early childhood offspring, body mass index, obesity

## Abstract

Preeclampsia (PE) is a major disease of pregnancy, with various short- or long-term complications for both the mother and offspring. We focused on the body mass index (BMI) of offspring and compared the incidence of obesity during early childhood between PE- and non-PE-affected pregnancies. Women with singleton births (*n* = 1,697,432) were identified from the Korea National Health Insurance database. The outcomes of offspring at 30–80 months of age were analyzed. The effects of PE on BMI and the incidence of obesity in the offspring were compared. The incidence of low birth weight (LBW) offspring was higher in the PE group (*n* = 29,710) than that in the non-PE group (*n* = 1,533,916) (24.70% vs. 3.33%, *p* < 0.01). However, BMI was significantly higher in the PE-affected offspring than that in non-PE-affected offspring. After adjusting for various factors, the risk of obesity was higher in the PE-affected offspring (odds ratio = 1.34, 95% confidence interval = 1.30–1.38). The BMI and incidence of obesity were higher during early childhood in the PE-affected offspring, even though the proportion of LBW was higher. These results may support the basic hypotheses for the occurrence of various cardiovascular and metabolic complications in PE-affected offspring. In addition, early-age incidence of obesity could influence PE management and child consultation in clinical applications.

## 1. Introduction

Preeclampsia (PE) is one of the most complex medical complications of pregnancy and is characterized by new-onset hypertension (HTN) and proteinuria after 20 weeks of gestation that affects approximately 2–8% of pregnant women worldwide [1,2]. In Korea, approximately 3% of pregnancies are affected by PE [3]. PE is associated with fetal growth restriction (FGR), placental abruption, preterm delivery, and an increased risk of maternal cardiovascular events later in life [4]. While much is known about the acute maternal and newborn complications in PE pregnancies, there have been few reports on the possible long-term effects [5].

PE has an adverse effect on the cardiovascular disease (CVD) risk profile in the later life of the mother [6,7]. Women who have experienced PE have approximately double the risk of CVD or cerebrovascular disease and three times the risk of HTN [5,6,7,8]. PE also reduces the life expectancy by up to 10 years; therefore, annual checkups for CVD and metabolic diseases are recommended after a PE pregnancy [8,9]. Some studies on the long-term health outcomes of offspring have found worsening of blood pressure, body mass index (BMI), diabetes mellitus (DM), and lipid profiles in adults born after a PE pregnancy [5,10].

Obesity is one of the main risk factors for adult CVD, and BMI is a diagnostic and evaluation tool for the development of obesity. Several meta-analyses have evaluated the relationship between BMI and CVD [11,12,13,14,15]. Moreover, childhood obesity may be a risk factor for selected adult CVD risk factors [16]. Further, the precursors of adult CVD begin in childhood, and pediatric obesity has an important influence on the overall CVD risk during the adult period [17]. These observations indicate the importance of evaluating BMI and its changes during early childhood.

Several studies have reported a relationship between PE pregnancy and cardiometabolic outcomes in PE-affected offspring. Overall, PE is not a problem of pregnancy alone, but rather a condition that alters long-term health outcomes for offspring, including negative effects on high blood pressure and BMI as well as potentially adverse health outcomes outside the cardiovascular system [5,10,18]. However, these recent studies mostly included young adults, not early childhood offspring. Therefore, although low birth weight (LBW) and BMI are common in PE-affected offspring, it is not clear how the BMI curve of PE-affected offspring crosses over the non-PE-affected offspring.

In this study, the BMI of offspring from 30 to 80 months of age was investigated in non-PE and PE pregnancies, to determine the risk of early childhood obesity in PE-affected offspring.

## 2. Materials and Methods

### 2.1. Health-Care Delivery System and Offspring in Korea

Almost 97% of the Korean population is enrolled in the Korea National Health Insurance (KNHI) program [19]. Therefore, the KNHI claims database contains information on all claims for approximately 50 million Koreans, and nearly all information about the extent of a disease can be obtained from this centralized database. As part of the KNHI system, a National Health Screening Program for Infants and Children (NHSP-IC) was started in 2007 and comprised seven consecutive health examinations based on age groups of 4–9, 9–18, 18–30, 30–42, 42–54, 54–66, and 66–80 months. The NHSP-IC data included information from physical examinations, anthropometric examinations, and developmental screenings. This study used the KNHI claims database to identify all women with singleton births between 1 January 2007, and 31 December 2010. We also determined their history of PE based on prepregnancy HTN or prepregnancy DM International Classification of Diseases (ICD) 10 codes (Appendix A). The diagnosis of PE includes the identification of high blood pressure (systolic BP at or above 140 mmHg, a diastolic BP at or above 90 mmHg, or both), proteinuria (≥300 mg per 24-h collection or >1+ on a urine dipstick), or the presence of elevated liver enzymes, high platelet count, headache, or visual disturbances after 20 weeks of gestation. Women were excluded from the analysis if their offspring had not undergone at least one of the seven consecutive NHSP-IC health examinations. The data of women who met the inclusion criteria were linked to the data of their offspring contained in the NHSP-IC database.

### 2.2. Data Set and Outcomes

Figure 1 shows a flowchart of the participant enrollment. Using the KNHI claims data extracted in January 2018. We identified all women with singleton births between 1 January 2007, and 31 December 2010, and then merged the data from the KNHI claims database and the NHSP-IC database. Datasets were excluded from the analysis if their offspring did not undergo at least one of the seven consecutive NHSP-IC health examinations or had missing data. Data on maternal and offspring outcomes were extracted. These included the history of prepregnancy HTN and DM; pregnancy outcomes such as parity, delivery mode, and PE; and neonatal outcomes such as neonatal sex, gestational age at birth, and birth weight. Preterm birth was defined as gestational age < 37 weeks, LBW was defined as birth weight < 2.5 kg, and large for gestational age was defined as birth weight > 4.0 kg. Offspring growth was assessed using BMI measurements that had been taken between 30 and 80 months of age. For this analysis, we used the results from the fourth, fifth, sixth, and seventh NHSP-IC screening examinations. Current BMI was categorized according to age- and sex-specific BMI values derived from the NHSP-IC. According to the NHSP-IC growth chart, underweight and obesity were defined as BMI ≤10th and ≥95th percentiles, respectively. To evaluate sex-affected obesity, male and female offspring were divided and evaluated. The study protocol was approved by the Institutional Review Board of the Korea University Medical Center (IRB Number: 2019GR0265).

### 2.3. Statistical Analysis

Continuous and categorical variables were expressed as mean ± standard deviation and number (percentage) values, respectively. Clinical characteristics were compared using the Mann–Whitney test for continuous variables and the Chi-square test for categorical variables. The risks of underweight and obese offspring relative to the presence or absence of a maternal history of PE were evaluated using multivariable logistic regression analysis at each follow-up period of NHSP-IC, using data from the fourth, fifth, sixth, and seventh screening examinations. Generalized estimating equations (GEEs) were used for the longitudinal evaluations of the association between a maternal history of PE and offspring growth by considering the correlation between repeated measurements in the same individual. All tests were two-sided, and statistical significance was set at *p* < 0.05. Statistical analyses were performed using SAS for Windows (version 9.4; SAS, Cary, NC, USA).

## 3. Results

### 3.1. Baseline Characteristics of the Study Population

The clinical characteristics of the patients in this study are shown in Table 1. Women with a PE pregnancy (*n* = 29,710) were older and showed significantly increased underlying HTN or DM compared to women without PE pregnancy (*n* = 1,533,916). In terms of obstetric characteristics, women with PE pregnancy showed higher rates of primiparity, cesarean section, and preterm birth than women without PE pregnancy. Neonates from the PE pregnancy group had a significantly lower average birth weight and a higher rate of LBW than women without PE.

### 3.2. BMI Distribution of Offspring between Women with or without a PE Pregnancy

Offspring growth was analyzed using BMI measurements made between 30 and 80 months of age in the fourth, fifth, sixth, and seventh NHSP-IC screening examinations (Table 2). BMI was lower until fourth period from PE affected offspring and did not differ significantly up to the fifth screening examination (30–42 months); however, from the sixth screening examination, it increased significantly in offspring from a PE pregnancy, and the gap between the two groups also increased over time. Female offspring showed the same pattern, while male offspring showed a significant increase from the 5th screening examination (Appendix A).

### 3.3. Prevalence and Associations between Maternal History of PE and Underweight or Obese Offspring

To evaluate the prevalence rates of obesity and underweight among offspring from a PE or non-PE pregnancy, all offspring BMI data were divided into obese or underweight groups (Table 3). The risk of obesity increased at all examinations in offspring from a PE pregnancy after adjusting for maternal age, maternal prepregnancy DM or HTN, primiparity, cesarean section, preterm birth, neonatal sex, and birth weight. The same result was observed when male and female offspring were categorized (Appendix A). However, no significant correlations were observed for the underweight offspring after adjustment (Table 3). When repeated underweight and obesity measurements at each follow-up period were considered using GEE, the risk of being underweight (odds ratio (OR) = 0.96, 95% confidence interval (CI) = 0.93–0.99 was decreased, but that of being obese (OR = 1.34, 95% CI = 1.30–1.38) was increased in the offspring of women with PE.

## 4. Discussion

In this study, we focused on the early childhood BMI of offspring and evaluated the prevalence of obesity after a PE pregnancy using nationwide data from Korea. This study found that despite having a lower birth weight, offspring from a PE pregnancy had a higher BMI as they aged compared to those from a non-PE pregnancy. There have also been other reports of offspring from PE pregnancies with an increased BMI. Davis et al. reported that the BMI was 0.62 kg/m^2^ higher in PE-affected young adults [10], while Thoulass et al. reported this as 0.44 kg/m^2^ higher [18]. In the present study, the BMI of PE-affected offspring significantly increased after the evaluation performed at 30–42 months of age, and the rate of BMI increase was also higher in the PE-affected offspring group. However, while the BMI gap between the two groups gradually increased, it remained lower than that in the two previous studies, at 0.05, 0.12, and 0.23 at 42–54, 54–66, and 66–80 months, respectively. This suggests that an increased BMI develops from early childhood, even in LBW offspring.

Our evaluations of the prevalence rates of obesity and underweight in offspring affected by PE pregnancy revealed that the risk of obesity increased in PE-affected offspring at all examination periods. Furthermore, the risk OR gradually increased as the offspring aged. The risk of obesity was 1.34-fold higher in PE-affected offspring after adjusting for other clinical factors and performing repeated measurements, whereas the underweight prevalence rate did not change significantly during each examination period. However, the adjusted OR values under GEE indicated that the risk of being underweight was lower in the offspring of women with PE. Therefore, future studies should evaluate the correlation between obesity in PE-affected offspring and the risk of CVD.

In some studies, sex-specific feto-maternal interactions affect significant biological mechanisms of genetics, epigenetics, and hormones in offspring [20]. Especially, a specific maternal race and female fetal sex increased the risk of PE pregnancy [21]. Therefore, male and female fetuses and neonates have different mechanisms to cope with adverse environments or events [22]. In this study, no differences in obesity were observed between men and women in early adulthood. In particular, in women, BMI was statistically different after 54 months; conversely, males showed differences after 42 months. However, after adjustment, the OR of obesity was similar in both men and women.

The Developmental Origin of Health and Disease hypothesis about the fetal origin of disease [23] was first proposed over 25 years ago, when Barker identified a relationship between birth weight and CVD in adults [24]. According to this hypothesis, a suboptimal intrauterine environment permanently alters organ structure and the function of biological feedback systems, thereby affecting individual susceptibility to diseases later in life [25]. The precise causes of PE development remain unknown. In a recent study, uteroplacental insufficiency based on abnormal placentation in early pregnancy and maternal response to abnormal maladaptation were suggested [26,27,28].

FGR is induced for various reasons, resulting in small for gestational age (SGA) offspring at birth. Therefore, postnatal catch-up growth occurred. In our study, increased BMI was similar to that found in SGA infants and in premature births [29,30]. In case of malnutrition, as one of the reasons for SGA, prenatal starvation is associated with epigenetic changes and effects for energy storage in adipose tissue and thus, overweight [31]. Offspring with postnatal catch-up growth have better insulin sensitivity, leading to a favorable growth and increased weight and BMI [32]. However, most SGA offspring catch up in weight and length before 12 months of age, [29] but this was not the case in our study subjects who experienced catch-up at a later age of approximately 42–54 months. In our opinion, the findings of our study support that FGR due to PE can lead to differences in SGA offspring due to other reasons.

This study has some limitations. The first was its retrospective design. PE can occur at different intensities at its onset, and some studies have suggested that early- and late-onset PE should be considered as different pathogenic conditions, thereby leading to different offspring outcomes [33]. The second limitation was the lack of adjustment for certain confounding factors, such as neonatal intensive care and exogenous glucocorticoids. A high rate of LBW among the PE-affected offspring induced a high exposure to perinatal interventions such as intravenous nutrition and mechanical ventilation, which can affect their growth [34]. Additionally, an increased probability of antenatal exposure to excessive glucocorticoids can affect the lipid profile or metabolic system of PE-affected offspring [35]. Furthermore, because of the lack of ICD-10 codes in the NHIS database, lack of adjusted gestational ages; birth weight percentile; first occurrence of PE; maternal prepregnancy status of BMI, HTN, diabetes, and other social status; and smoking or alcohol habits may be confounding factors. Additionally, maternal weight gain and newborn nutrition, including breastfeeding and formula, may have affected the results but were not included in the factors. In particular, although Davis et al. showed that prepregnancy BMI is a crucial factor of PE-affected offspring cardiovascular outcome in a systemic review, it was not included in this study [10]. In Korea, a recent study reported a mild increase of prepregnancy BMI in PE-affected mothers than that in non-PE-affected mothers (21.05 ± 2.85 (*n* = 3391) vs. 20.48 ± 2.44 (*n* = 45,674)) [36]. Therefore, we believe our study pool was similar to this report, because the NHIS enrolled Koreans of the same ethnicity. Although there is a difference in prepregnancy BMI between PE and non-PE pregnancy, both almost had normal weight ranges (BMI 18.5–24.9). Therefore, adjustment of the effect of pre-pregnancy BMI effect may not be serious. Although adjusting preterm birth, as a natural characteristic of PE, including preterm birth, has direct effects on PE-affected offspring growth, further studies are still needed. To overcome the limitations of this study, further prospective studies will be necessary for subgroup analysis using medical resuscitation of neonates and maternal prepregnancy conditions such as BMI, HTN, or DM. Third, although PE is a pregnancy complication that worsens the intrauterine environment, not using gestational age-adjusted birth weight may be a confounding factor closely related to offspring obesity.

Notwithstanding the above limitations, our study had the advantage of involving a large nationwide assessment of the association between maternal PE history and almost seven years of follow-up of offspring growth. To the best of our knowledge, this is the first large longitudinal study of the early childhood period. A few previous studies found no difference among children younger than 10 years, but they involved small PE-affected populations (*n* = 23) [37]. Mogren et al. reported a similar OR for obesity in PE-affected offspring as in the present study, but their analysis was limited to adults aged 29–41 years [38]. Therefore, our study may contribute to the understanding of the impact of maternal PE on long-term offspring health and the pathophysiology of PE. Therefore, evaluation of BMI of PE-affected offspring in early childhood, as well as maternal evaluation of HTN or diabetes, is important to manage and prevent PE pregnancy.

In conclusion, the risk of early childhood obesity is increased in pregnancies complicated by PE. Together with further research, the results are likely to be useful to consult PE pregnant mothers and their offspring about early childhood management.

## Figures and Tables

**Figure 1 jcm-10-03758-f001:**
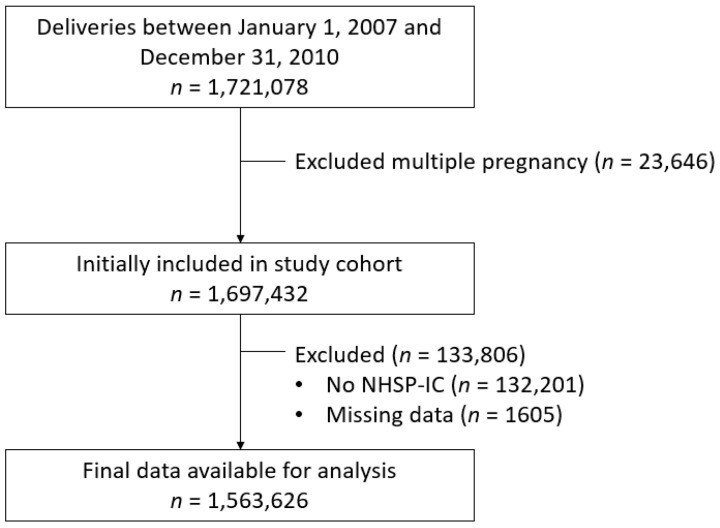
Flowchart of participant enrollment. NHSP-IC, National Health Screening Program for Infants and Children.

**Table 1 jcm-10-03758-t001:** Prepregnancy and pregnancy characteristics stratified according to maternal preeclampsia (PE).

Characteristic	Women without PE (*n* = 1,533,916)	Women with PE (*n* = 29,710)	*p*
Age, years	30 (28–30)	31 (28–34)	<0.0001
Advanced age, >35 years	213,048 (13.89%)	6026 (20.28%)	<0.0001
Prepregnancy HTN	41,363 (2.70%)	3312 (11.15%)	<0.0001
Prepregnancy DM	59,882 (3.90%)	2172 (7.31%)	<0.0001
Primiparity	804,510 (52.45%)	19,805 (66.66%)	<0.0001
Cesarean section	535,393 (34.90%)	17,615 (59.29%)	<0.0001
Preterm birth	36,567 (2.38%)	5636 (18.97%)	<0.0001
Neonatal sex, male	792,136 (51.64%)	14,841 (49.95%)	<0.0001
Birth weight, kg	3.2 (3–3.5)	3.0 (2.5–3.3)	<0.0001
LBW	51,104 (3.33%)	7337 (24.70%)	<0.0001
LGA	62,590 (4.08%)	1077 (3.63%)	<0.0001

Data are presented as medians with interquartile ranges or *n* (%) values. Abbreviations: HTN, hypertension; DM, diabetes mellitus; LBW, low birth weight; LGA, large for gestational age.

**Table 2 jcm-10-03758-t002:** Body mass index of offspring up to 80 months of age stratified according to maternal preeclampsia (PE).

Age	Offspring Born to Women without PE	Offspring Born to Women with PE	*p*
30–42 months	16.01 (15.19–16.88)	15.97 (15.12–16.89)	0.0009
42–54 months	15.91 (15.12–16.77)	15.93 (15.08–16.85)	0.4419
54–66 months	15.79 (14.98–16.74)	15.86 (14.97–16.89)	<0.0001
66–80 months	15.80 (14.90–16.93)	15.95 (14.95–17.28)	<0.0001

Data are presented as median with interquartile range values.

**Table 3 jcm-10-03758-t003:** Prevalence and associations between maternal history of preeclampsia (PE) and underweight and obese offspring until 80 months of age.

Age	Offspring Born to Women without PE	Offspring Born to Women with PE	*p*	Unadjusted OR (95% CI)	Adjusted * OR (95% CI)
Obesity					
30–40 months	77,702/984,992 (7.8%)	1741/19,256 (9.0%)	<0.001	1.16 (1.10–1.22)	1.28 (1.21–1.34)
42–54 months	113,298/958,503 (11.8%)	2703/19,043 (14.1%)	<0.001	1.23 (1.18–1.29)	1.34 (1.28–1.39)
54–66 months	111,415/890,444 (12.5%)	2838/17,886 (15.9%)	<0.001	1.32 (1.27–1.37)	1.36 (1.31–1.42)
66–80 months	81,917/626,585 (13.1%)	2178/12,404 (17.6%)	<0.001	1.42 (1.35–1.49)	1.42 (1.35–1.40)
Underweight					
30–40 months	97,443/984,992 (9.8%)	2220/19,256 (11.5%)	<0.001	1.19 (1.14–1.24)	0.96 (0.92–1.01)
42–54 months	69,920/958,503 (7.3%)	1718/19,043 (9.0%)	<0.001	1.26 (1.20–1.33)	1.01 (0.95–1.06)
54–66 months	65,352/890,444 (7.3%)	1491/17,886 (8.3%)	<0.001	1.15 (1.09–1.21)	0.94 (0.89–0.99)
66–80 months	51,884/626,585 (8.3%)	1171/12,404 (9.4%)	<0.001	1.16 (1.09–1.23)	0.97 (0.91–1.03)

Data are *n* (%) values, except where indicated otherwise. * Adjusted for maternal age, maternal prepregnancy DM, primiparity, cesarean section, preterm birth, neonatal sex, and birth weight. Abbreviations: OR, odds ratio; CI, confidence interval.

## Data Availability

The data presented in this study are openly available upon reasonable request.

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
