# Peer review of "Risk of Early Childhood Obesity in Offspring of Women with Preeclampsia: A Population-Based Study"

_jcm, 2021, doi:10.3390/jcm10163758_

Round 1

Reviewer 1 Report

This reviewer would like to appreciate the authors who fulfilled most of my suggestions. Overall, the manuscript looked more readable. However this reviewer still remained unsatisfied with a couple of points which are shown as followings:

  1. It seems that the p-value (0.0009) from the age category “42-54 months” in Table 2 appeared too much deviated from those shown for the same category in Supplementary table 2-3, (0.0367 and 0.5986). As Table 2 is a merger of Supplementary table 2-3, it is hard to believe observing such a low p-value, given those in Supplementary tables. Also, for those tables, authors might like to keep the same decimal point (up to hundredths e.g. 16.10) as they did in the previous version. Currently shown as 15.9  for both PE and non-PE with p-value of 0.0009.
  2. Line 47, “The some studies” => “Some studies”

Author Response

  • We especially appreciate your thoughtful suggestions and insights. The manuscript has benefited from these insightful suggestions. We prepare the answer for your suggestion in the newly revised manuscript.
  • "Please see the attachment." 

Reviewer 2 Report

The manuscript requires a substantial English language editing before considering this paper for revision and publication.

Author Response

First of all, we are sorry about insufficient English proofreading. For your suggestion, we did a newly revised process of English (Editing by Editage, code OKDDI_ 5). We attached a certificate of this process.

Round 2

Reviewer 2 Report

no further comments 

This manuscript is a resubmission of an earlier submission. The following is a list of the peer review reports and author responses from that submission.

Round 1

Reviewer 1 Report

The manuscript of Yang et al, submitted for publication at Journal of Clinical Medicine, studied the Body Mass Index (BMI) of offspring from the women who were diagnosed with pre-eclampsia (PE) during their pregnancy and compared that of offspring whose mothers were non-PE pregnancies in
Korean population. They reported that the BMI of children from PE- pregnancies was significantly higher than that of offspring from non-PE cases and concluded that the risk of early childhood obesity is increased in pregnancies complicated by PE. Their retrospective study is based on a sample size of over a million, which appears to add more statistical power. However, large cohort size can be presumably very heterogeneous (e.g PE (sub)phenotypes, mixture of early/late onset PE cases, maternal BMI ranges, and neonatial sex) and that could possibly hamper their analysis -
these limitations were already well-described in Discussion. That said, this reviewer thought, at least they could have taken the sex of offspring into account to avoid its potential confounding effect on BMI (see below for detail). Overall, the conclusion of Yang et al. does not seem to contradict with other previous reports. The current report can be an example showing how the use of electronic health records (or something with that nature) can be utilized in a large longitudinal cohort study and what should be taken into account for a better setup/design of future studies.

Major comments
1. There are growing evidence suggesting the correlation between feto-placental sex and adverse pregnancy outcomes such as PE, e.g Al-Qaraghouli and Fang 2017, Clifton 2010, Gong et al 2018, Mirzakhani and Weiss 2020. This seems to be reflected in Table 1 - male neonates being more frequent in non-PE pregnancy than PE-affected mothers. This reviewer would like to suggest analysing Table 2 and Table 3 by the sex of offspring, although acknowledging that the neonatal sex was adjusted togather with other variables in Table3. The authors might want to discuss the sex effect of offspring on BMI in PE and non-PE pregnancies, depending on their new analysis.

2. Considering the definition of childhood obesity (≥95th percentiles, as shown line 101-102), why does the percentage of being in that category appear higher? For example, at the 7th screening examination (i.e. 66-80 months) there are 84,095 (81,917+2,178) obese offspring out of 638,989 (626,585+12,404) pregnancies that were examined - that is 13.2%. At the same 7th screening (66-80 months), the percentage of underweight offspring is 8.3% (i.e. (51,884+1,171)/(626,585+12,404)) which is within the threshold of underweight (i.e. ≤10th percentile).

3. What is the supporting statistical test for the following statement: “However, there were no significant correlations for the underweight offspring”, shown line 148-149. Does it refer to adjusted or unadjusted odd ratio?

4. This reviewer found some unusual approaches in their data processing and analysis where this study is based upon. Firstly they summarized various measurements using the arithmetic mean with standard deviation as shown in Table 1-3. As the mean could be affected by outliers especially in a large sample size, the authors might want to consider using the median with interquartile range (IQR). Secondly, this reviewer wondered why the authors chose to use t-test over nonparametric alternatives such as Mann-Whitney U test (MWU). It would be reassuring if the same level of statistical significance could be achieved with MWU.

5. If the journal’s formatting allows, is it possible to present exact p-values (i.e. P=1.5x10-5)?

6. Line 82, please explain “ICD-10” at its first appearance and list which specific codes were used with regard to the authors comment at line 211 “not available ICD-10 coding in NHIS database”.

Minor comments
1. The title does not read well and it is grammatically not correct: “When is the time of obesity evaluation begin in offspring from preeclampsia pregnancy?”.

2. The reference numbers were placed after full stop marks (e.g. line 38 and 39) - authors need to correct these.

3. The use of “thousand-separator”, i.e. “,” is not consistent in Table 1.

4. The authors might want to have their manuscript proof-read to correct some grammatical errors. See below for examples (but not limited):
a. Line 48, “The few studies” -> “Some studies” or “A few studies”
b. Line 53, remove space before the full stop mark.
c. Line 60, “pre-eclampsia” -> “preeclampsia” or “PE” for consistency.
d. Line 64, “being limited” -> “limited”.
e. Line 136, “up to 30-42 months” => “up to the 4th screening examination (30-42 months)”.
f. Line 199-201 - it did not read well.
g. Place a full stop at line 217 - e.g “[34]. Therefore".
h. Line 221, “is might not be” -> “might not be”.

Reviewer 2 Report

The authors presented a population based study focused on the early childhood BMI of offspring, comparing the incidence of obesity during the early childhood period between PE and non-PE affected pregnancy.

They found that despite having a lower birthweight, offspring from a PE pregnancy had a higher BMI as they aged compared to those from a non-PE pregnancy.

The study has major issues, particularly in the study design that may affect the results:

  1. The definition of PE is not described in the study
  2. LBW and LGA are defined based on < 2.5 and >4.0 Kg and not related to GA or defined based on percentiles: a significant higher rate of PTB was found in the PE group and this should be considered directly affecting birthweight, particularly for LBW.
  3. Although acknowledged in the limitations, there are many possible confounding factors such as maternal BMI, rate of GDM, maternal weight gain, newborn nutrition (breastfeeding or formula) that are not available and that may have a significant impact on the final results.